# Effects of Physical Activity on the Stress and Suicidal Ideation in Korean Adult Women with Depressive Disorder

**DOI:** 10.3390/ijerph17103502

**Published:** 2020-05-17

**Authors:** Kyo-Man Koo, Kyungjin Kim

**Affiliations:** 1Department of Adapted Physical Education, Baekseok University, Cheonan-Si 31065, Korea; nicekm73@hanmail.net; 2Department of Adapted Physical Education, Korea National Sport University, Seoul 05541, Korea

**Keywords:** physical activity, stress, suicidal ideation, Korean adult women, depressive disorder

## Abstract

Depressive disorder is common in many adult women in the world. It was found that depressive disorder was related to stress and suicidal ideation in Korean adult women with depressive disorder. Physical activities were effective to solve this mental disorder. Thus, the purpose of the study was to investigate the effects of physical activity (PA) on the stress and suicidal ideation of Korean adult women with depressive disorder. A sample of 1315 Korean adult women who have depressive disorder was collected, and the Korea National Health and Nutrition Examination Survey (KNHANES) was used to determine this purpose of the study. The complex samples in frequency analysis were used to identify the characteristics of the participants. The logistic regression in the complex sample design was conducted to investigate the effects of PA on the stress and suicidal ideation in Korean adult women with depressive disorder. The effects of flexibility exercise on the stress in Korean adult women with depressive disorder showed the odds ratio (OR) value of 1.434 (OR = 1.434, 95% CI = 1.043–1.973, *p* < 0.05). The effects of flexibility exercise on the suicidal ideation in Korean adult women with depressive disorder presented the OR value of 0.682 (OR = 0.682, 95% CI = 0.496–0.937, *p* < 0.05). Based on the results, the participants who did flexibility exercises were likely to have less stress and suicidal ideation than the Korean adult women with depressive disorder who did not participate in flexibility exercise. In conclusion, the flexibility exercise has played an important role in reducing and preventing stress and suicidal ideation in Korean adult women with depressive disorder.

## 1. Introduction

Depressive disorder is a serious mental disorder in the world. It is known that depressive disorder has negative influences on how a person feels, thinks, and behaves. It can cause a variety of physical and emotional troubles, such as feeling sad, obesity, loss of interest, sleeping, slowed movements, purposeless physical activity, and thoughts of suicide [1]. According to the World Health Organization (WHO), more than 350 million individuals have depressive symptoms across the world. In addition, depressive disorder is predicted to be the second most significant community health problem after 2020 [2]. The WHO stated that people who have depressive disorder need more consideration and care. It is a more frequent mental disorder in women than in men all over the world. Hormonal changes, pregnancy, childbearing, and menopause are reasons why adult women suffer from depressive disorder more than adult men [3]. These transitions can affect mental health in many adult women across the world. For instance, many adult women have experienced anxiety and depression symptoms during pregnancy [4]. Further, adult women may have less free time than adult men due to the social expectation of caring for their children.

In South Korea, over 15% of people have experienced major depressive disorder during their lifetime [5,6]. In addition, around 25% of adult women have suffered from mental issues and depression during their lifetime [5]. According to the Korea Ministry of Health and Welfare [6], Korean adult women are approximately twice as likely as adult men to experience associated depressive symptoms. The life expectancy difference, due to low social and economic level, sedentary life, pregnancy, childbirth, and menopause between Korean women and men is the major reason that depressive disorder is more prevalent in Korean adult women [7]. These data imply that depressive disorder is becoming the most prevalent mental concern among Korean adult women. 

Particularly, depressive disorder is common in adult women. Many adult women are under pressure that affects life stress. This life stress has been reported to be highly correlated with depression [8]. Further, many biological, psychological, and social factors can contribute to depression. These conditions can interfere with their everyday activities, which may cause some depression symptoms related to suicidal ideation [2,3]. When stress affects suicidal ideation, it is argued that depression has a mediating effect [9,10]. The suicide rate among young adult people aged 20–29 was 25.6% of the death rate, the highest cause of death. In fact, it was 40% more than the mortality rate from traffic accidents in South Korea [11]. Many young adult people are in the process of transition to adult society and have complex stresses such as family expectation and social pressure on economic independence through employment [12]. The recent socioeconomic difficulties have led to suicidal ideation because of the increase in young adult people’s unemployment rate and heavy work-related stress in South Korea. Especially, young adult women are under more stress as their employment rate is only about 70% of that of men [11]. 

Especially, Korean adult women who have depressive disorder are more likely to attempt drug abuse or alcohol abuse, and over 10% of Korean adult women were reported to attempt suicide or have suicidal ideation [7]. Although many Korean adult women who have depressive disorders need to receive professional help, they hesitate to seek a specialist and be treated for it because of the mindset of Koreans having a mental disorder as bringing shame to the family [13]. Hence, suicide of Korean adult women is a serious social concern in South Korea, as the suicide rate has been the highest of the Organization for Economic Co-operation and Development (OECD) countries in the last decade [6].

Depressive disorder is caused by a variety of factors and increases the suicidal ideation of women [1,2,3]. In fact, 40% to 80% of the elderly who tried to kill themselves were reported to have depression in the United States [14]. Further, depression had a direct impact on the suicide risk of American young people [15]. Many studies have been conducted to solve such problems. Researchers provide evidence that regular physical activity (PA) has positive effects on the progressive processes of depressive disorder and mental health [16,17]. Several studies found that PA assists to prevent depressive symptoms of adult women and men [18,19]. Researchers have found that participation in PA helped to reduce depressive symptoms for adult women [20,21]. Robledo-Colonia et al. [21] stated that PA (e.g., aerobic exercise training) reduced depressive symptoms during pregnancy. Pereira et al. [20] reported that PA contributed to reduced weight and depressive disorder scores for postpartum women. Additionally, PA was found to be effective in the prevention of stress, depressive disorder, and suicidal ideation in girl students and elderly women. Participation in PA can positively cope with negative stress affecting mental health, including depression symptoms as thoughts of suicide [22,23].

However, many researchers have focused on reducing or preventing depressive symptoms [24,25,26] or the suicide of girl students and elderly females [22,23]. Several intervention studies for depressive disorder determined that the effect of only one kind of PA was analyzed rather than various PA types. Chu et al. [27] reported the effects of yoga on depressive symptoms in women, and Bernard et al. [28] found the effects of walking on depression in post-menopausal women. Many studies also have limited participants to examining the effects of PA intervention on women with depressive disorder. Chu et al. [27] provided evidence for the effects of PA on depressive symptoms in an intervention group (e.g., *n* = 13), and LeCheminant et al. [29] identified the effects of PA on depressive disorder in an intervention group (e.g., *n* = 30). Furthermore, there was limited research to examine stress and suicidal ideation of adult women with depressive disorder [30,31] even though these factors cause a serious social concern.

This study determined the effects of PA on the stress and suicidal ideation in Korean adult women with depressive disorder by using the Korea National Health and Nutrition Examination Survey (KNHANES), which was conducted by the recommendations of the WHO [32]. It has been used to reveal differences in perceived health status in national and demographic surveys targeting the general population (aged 16–65) in South Korea. In addition, these data can be used to solve the limitation about the number of participants and PA types. Therefore, the purpose of the study was to investigate the effects of PA (e.g., strength exercise, flexibility exercise, walking) on the stress and suicidal ideation of Korean adult women with depressive disorder.

## 2. Materials and Methods

### 2.1. Participants

The purpose of the study was to determine the effects of PA (e.g., strength exercise, flexibility exercise, walking) on the stress and suicidal ideation in Korean adult women with depressive disorder. This study used the data of KNHANES, which was conducted after the approval of the institutional review board (IRB) in Korea Centers for Disease Control and Prevention. In addition, it is represented as statistical data in South Korea, which is extracted every year by stratified, clustered, and systematic sampling. The KNHANES was conducted six times from 1988 to 2015.

The Korea Center for Disease Control and Prevention recommends that this raw data be reflected in the complex sampling because the sample design of the KNHANES was extracted using a two-stage stratified cluster sampling rather than a simple random sampling. The sampling framework of the KNHANES used the recent population, which is possible to extract a representative sample for the population over 19 years old living in South Korea. When using weights in the process of analyzing the data of the KNHANES, the inclusion errors, inequality extraction rates, and non-response errors of the surveyed participants were corrected according to the difference in the number of households and populations between the sample design time and the time of the survey. It was made to improve the accuracy of estimates related to the prevalence of chronic diseases, food, and nutrition. For instance, the population mean estimate ($\frac{\wedge }{Y}$) of the item of interest (y) is calculated as a weighted sample average by reflecting the weight ($w_{i},\;i=1,\;\ldots ,\;n$) to a total of survey data ($y_{i},\;,\;i=1,\;\ldots ,\;n$) of *n*. The weighted sample average is calculated as follows
(1)$$\frac{\wedge }{Y}=\overset{n}{\underset{i=1}{\sum }}w_{i}y_{i}/\overset{n}{\underset{i=1}{\sum }}w_{i}$$

The types of weights in the KNHANES were largely classified as household weights for analysis of household units and personal weights for analysis of individual units, and the household weights were given to survey households to represent the entire household in Korea and individuals to represent the entire population of the country. Individual weights were also divided into separate weights for each factor segment due to differences in the number of participants in each factor.

In this study, the fourth, fifth, and sixth KNHANES data from 2007 to 2015 were merged vertically to increase the statistical validity. The data of the fourth KNHANES from 2007 to 2009 included 24,871 participants, and the fifth KNHANES data from 2010 to 2012 involved 31,596 participants. The data of the sixth KNHANES from 2013 to 2015 included 22,948 participants. In this study, the study sample from a total of 1315 respondents (unweighted cases, women of over 19 years old) who completed questions in the KNHANES related depressive disorder were collected for the analysis. In this study, 1315 respondents have been maintained at least two weeks for a diagnosis of depression and diagnosed with depression by a specialist. The rest of the raw data were removed because it did not address questions in relation to depressive disorder. Further, the data collected from only common questions reduced the number of raw data since each KNHANES (e.g., the fourth, fifth, sixth KNHANES) contained several different questions in PA and the depressive disorder category. In Table 1, there were demographic characteristics of participants such as age, economic activity, educational level, household income, and residential area. In addition, age, activity restriction, subjective body image, and subjective health assessment were adjusted variables that impact on stress and suicidal ideation in Korean adult women with depressive disorder, except for PA in the KNHANES.

### 2.2. Research Instrument

The KNHANES was based on the International Physical Activity Questionnaire (IPAQ), which is a standardized questionnaire designed to measure and compare the level of PA of various populations (aged 16–65) around the world. In South Korea, the KNHANES has begun to collect data of the health level of people and evaluate the national health policy for the national health promotion project. The health behaviors of smoking, drinking, nutrition, and PA among the KNHANES contents are very important to prevent the health risk factors of Korean people. PA is the most cost-effective and highly efficient way of practicing health care for the prevention of chronic diseases, such as heart disease, hypertension, stroke, diabetes, colon cancer, breast cancer, and depression [33,34]. The KNHANES has been conducted to identify the health level of the Korean people and establish health policy since 1988.

In the IPAQ, specific examples for each country were used, and cultural diversity was considered to help understand the correct definition of PA [35]. There is a long-form questionnaire and short-form questionnaire in the IPAQ [36]. The Korean version (KNHANES) of short form of IPAQ was used in the national survey. According to Craig et al. [35], they conducted the validation of the short form of IPAQ, which calculated the Pearson correlation coefficient ranging from 1.12 to 0.57. Further, a study [37] investigated the correlation between accelerometer and the Korean version (KNHANES) of IPAQ. As a result, the Spearman correlation coefficient between the Korean version (KNHANES) of IPAQ and the PA measured by the accelerometer was 0.27.

### 2.3. Research Variables

The variables of this study were selected as type of PA, stress, and suicidal ideation for 1 year, and variables were categorized according to the purpose of this study by using the National Health and Nutrition Examination Guidelines [38]. 

First, physical activity, which is an independent variable, was set as strength exercise, flexibility exercise, and walking. In the walking variable, there was a question, which was, “How many days did you walk at least 10 minutes at a time in the last week?” Further, a 7-point scale (i.e., 1 = Never, 2 = 1 day, 3 = 2 days, 4 = 3 days, 5 = 4 days, 6 = 5 days, 7 = 6 days, 8 = everyday) was added to the walking question. These raw data were categorized as “1 = Never and 2 = Participation in walking” for this study. In the strength exercise variable, there was a question, which was, “How many days did you perform strength exercises such as push-up, sit-up, dumbbell, weight, and iron bar in the last week?” A 6-point scale (i.e., 1 = Never, 2 = 1 day, 3 = 2 days, 4 = 3 days, 5 = 4 days, 6 = 5 days and over) was used to the strength exercise question. The raw data of the strength exercise variable were classified as “1 = Never and 2 = Participation in strength exercise” for the study. In the flexibility exercise variable, there was a question, which was, “How many days did you do flexibility exercises such as stretching, bare-hand exercise, etc. in the last week?” A 6-point scale (i.e., 1 = Never, 2 = 1 day, 3 = 2 days, 4 = 3 days, 5 = 4 days, 6 = 5 days and over) was added to the flexibility exercise question. These raw data were categorized as “1 = Never and 2 = Participation in flexibility exercise.”

The perception of stress, which is a dependent variable, was asked as “How much stress do you usually feel in your daily life?” A 2-point scale (i.e., 1 = Low, 2 = High) was used for the perception of stress question. These raw data were used without modification. In the suicidal ideation variable, there was a question, “During the past 12 months, have you ever seriously thought about committing suicide?” A 2-point scale (i.e., 1 = Yes, 2 = No) was added to the suicidal ideation question. The raw data of suicidal ideation variable were used without change. In the process of restructuring the selected research variables, it was used for the analysis through review of previous studies and consultation among researchers.

### 2.4. Data Analysis

The purpose of this study was to investigate the effects of PA (e.g., strength exercise, flexibility exercise, walking) on stress and suicidal ideation in Korean adult women with depressive disorder. In order to achieve the purpose of the study, researchers reflected the design of the complex sample recommended in the National Health and Nutrition Examination Guidelines [38]. The stratification variable was assigned to the dispersion estimation layer, and the survey area was assigned to the cluster after the data from 2007 to 2015 were vertically combined. As the sample weight was integrated from 2007 to 2015, the integrated weights were calculated to apply to the analysis and the plan files were generated for the analysis. The integration rate was 0.5/8.5 in 2007 and 1/8.5 from 2008 to 2015.

The complex samples in frequency analysis were used to identify the characteristics of the participants. Furthermore, the logistic regression in the complex sampling design was conducted to investigate the effects of PA on the stress and suicidal ideation in Korean adult women with depressive disorder. The significance level was *p* < 0.05. After the user-missing value was changed to ‘valid values’, all data with the missing data were included for analysis, which was used to prevent the occurrence of bias in the dispersion estimation and the dispersion estimator. The SPSS 21.0 statistical package was used for all statistical analyzes.

## 3. Results

### 3.1. Effects of Physical Activity (e.g., Strength Exercise, Flexibility Exercise, Walking) on the Stress in Korean Adult Women with Depressive Disorder

Logistic regression analysis was used to determine the effects of PA (e.g., strength exercise, flexibility exercise, walking) on stress in Korean adult women with depressive disorder. As shown in Table 2, the stress of Korean adult women with depressive disorder was a significant correlation in flexibility exercise. However, there were no statistically significant differences in strength exercise and walking. The Korean adult women with depressive disorder who participated in flexibility exercise showed the odds ratio (OR) value of 1.434 (OR = 1.434, 95% CI = 1.043–1.973, *p* < 0.05). Therefore, Korean adult women were less likely to be stressed when participating in flexibility exercise.

### 3.2. Effects of Physical Activity (e.g., Strength exercise, Flexibility Exercise, Walking) on the Suicidal Ideation in Korean Adult Women with Depressive Disorder

Logistic regression analysis was used to identify the effects of PA (e.g., strength exercise, flexibility exercise, walking) on suicidal ideation in Korean adult women with depressive disorder. In Table 3, the flexibility exercise had a significant correlation on suicidal ideation in Korean adult women with depressive disorder. On the other hand, the strength exercise and walking did not have statistically significant differences. The Korean adult women with depressive disorder who participated in flexibility exercise presented the OR value of 0.682 (OR = 0.682, 95% CI = 0.496–0.937, *p* < 0.05). Therefore, Korean adult women with depressive disorder had less suicidal ideation when participating in flexibility exercise.

## 4. Discussion

The purpose of the present study was to determine the effects of PA (e.g., strength exercise, flexibility exercise, walking) on the stress and suicidal ideation in Korean adult women with depressive disorder. The logistic regression in the complex sample design was conducted to determine the purpose of the current study. Based on the findings, Korean adult women with depressive disorder who participated in flexibility exercise had less stress and suicidal ideation than those who did not. However, there were not significant effects of strength exercise and walking on stress and suicidal ideation in Korean adult women with depressive disorder.

Adult women with depressive disorder might be associated with low physical condition and a sedentary lifestyle [39]. Galper, Trivedi, Barlow, Dunn, and Kampert [40] suggested that physically active people have significantly lower depression levels than inactive people. Correspondingly, PA may be important to prevent depressive symptoms and for behavioral interventions to reduce depression among adult women [41]. Further, PA might be one method without pharmacological action and be beneficial in decreasing the stress and suicidal ideation in adult women with depressive disorder [42]. 

In the current study, flexibility exercise contributed to reducing the stress and suicidal ideation in Korean adult women with depressive disorder. These results were related to findings by Esgalhado et al. [42], who reported that flexibility exercise (i.e., stretching exercise) decreased musculoskeletal pain, fatigue, and anxiety. In addition, it improved the quality of nutrition, sleep, and rest. According to Vergeer and Roberts [43], the flexibility exercise helped to feel the body image which promotes the participants to see and to feel muscles. These experiences made them comfortable and relaxed. Increases in PA in flexibility exercise have been helpful to reduce negative thoughts, eliminate worries, improve social relationships, and alter brain chemistry to have a positive mind [44]. Physical fitness improvement and regular PA in flexibility exercise also decreased the anxiety symptoms [45]. These factors, such as reducing pain, anxiety, fatigue, negative thoughts, worries, improving social connection and the quality of nutrition, sleep, and relaxation, might have a significant impact on mental, physical, and social health of adult women with depressive disorder.

Furthermore, many researchers [27,46,47] showed significant reductions in depressive symptoms and stress after the flexibility exercise. Tamil Selvi and Thangarajathi [46] reported that the flexibility exercise was effective to reduce depression and stress. Flexibility exercise can provide a positive effect on mental, physical, and social well-being for adult women with depressive disorder. For instance, flexibility exercise can block negative feelings because it can increase the mood by concentrating the interest on the way that the feelings are conveyed in the body [27,46]. Korean adult women with depressive disorder can be aware of the connections between the mental and physical levels while performing flexibility exercise. This experience may improve mental health for them. Especially, mental health is linked to depressive disorder and stress in the community. The depressive disorder and stress were showed to be related to increased risk of suicidal ideation [48]. These results were related to findings by Bantjes, Kagee, Mcgowan, and Steel [49], who reported that stress might contribute to poor mental health and precipitate suicidal ideation. These researchers stated that the evidence-based interventions are necessary to prevent stress and suicidal ideation in depressive disorder. 

On the other hand, there were not significant effects of strength exercise and walking on stress and suicidal ideation in Korean adult women with depressive disorder. Even though there is no research investigating the effects of strength exercise and walking on stress and suicidal ideation in Korean adult women with depressive disorder, it might be deduced from several facts. The fifth and sixth KNHANES did not ask questions about moderate intensity strength exercise and walking. However, many Korean adults are doing moderate intensity strength exercise and walking. According to the Korea Ministry of Culture, Sports and Tourism [50], 44.7% Korean women walked at a moderate intensity. The questions in the KNHANES could lead to these results. In addition, participants in the current study have suffered from depressive disorder, which is more likely to be a negative emotional state than a positive emotional state. The state of negative emotions might affect mental health in Korean adult women with depressive disorder, which is associated with stress and suicidal ideation [48]. Müller, Hess, and Hager [51] reported that the quality of life of patients was found to be strongly influenced by depressive disorder rather than the PA. Furthermore, PA is initially related to stress from learning exercise skills, facing challenges, and getting frustrations [52]. Incorrect setting of exercise intensity also might be a reason for stress from PA. These factors may not help to provide positive feelings for Korean adult women with depressive disorder. Thus, it might lead Korean adult women with depressive disorder to see negative effects of strength exercise and walking on stress and suicidal ideation.

## 5. Research Limitations

In the study, there were some limitations. First, the results of the current study may not be generalizable to adult women with depressive disorder in other countries because many studies presented were related to South Korea. In addition, the participants were recruited in South Korea; therefore, the generalization of the findings should be restricted to Korean adult women with depressive disorder due to cultural differences. Future studies should consider providing evidence that can be generalized to other countries. Another limitation was that the KNHANES did not use structured and validated scales to determine stress and suicidal ideation. This may present an important limit to the validity and reliability of the KNHANES in stress and suicidal ideation. Future KNHANES should consider determining structured and validated scales for stress and suicidal ideation. The third limitation was that the study did not include a comparison sample of Korean adult women without depressive disorder because the difference in the number of samples between the two groups (e.g., Korean adult women with and without depressive disorder) was too big. Future research should include a comparison sample to determine the effects of PA on the stress and suicidal ideation in Korean adult women with and without depressive disorder. The fourth limitation was that this study showed a lot of risk factors for suicidal ideation in Korean adult women with depressive disorder (e.g., severity of depression, pharmacological treatment, mixed features). However, these risk factors were not analyzed in the present study. Hence, this study did not present ways to address risk factors for suicidal ideation in Korean adult women with depressive disorder. Future research should suggest ways to deal with risk factors for suicidal ideation in Korean adult women with depressive disorder. The final limitation was that the KNHANES did not show the examples of events by intensity of PA, and explanations of PA intensity were not appropriate. It can cause errors in the rate of PA by intensity. Future KNHANES items in PA need to be improved to meet the purpose of the survey and be understandable to respondents.

## 6. Conclusions

Depressive disorder is common in many adult women in the world. It is linked to stress and suicidal ideation in Korean adult women with depressive disorder. PA is an effective way to resolve this mental disorder. The present study aimed to determine the effects of PA (e.g., strength exercise, flexibility exercise, walking) on stress and suicidal ideation in Korean adult women with depressive disorder. The conclusions of this study were as follows: in this study, flexibility exercise played an important role in reducing and preventing stress and suicidal ideation in Korean adult women with depressive disorder. However, strength exercise and walking did not have significant effects on stress and suicidal ideation in Korean adult women with depressive disorder. Future studies need to consider determining which exercises aside from strength exercise, flexibility exercise, and walking are effective to reduce stress and suicidal ideation in women with depressive disorder.

## Figures and Tables

**Table 1 ijerph-17-03502-t001:** Demographic characteristics of participants and adjusted variables.

Characteristics		Unweighted *N*	Weighted *N*	% (Weighted)
**Age**	19–29	74	66,517.6	11.3
	30–39	164	108,134.5	9.4
	40–49	187	142,869.9	14.2
	50–59	274	161,737.7	23.2
	60–69	331	134,858.6	19.6
	70–79	250	101,351.0	17.5
	80≤	35	18,415.5	4.9
Economic activity	Employment	439	271,010.1	37.0
	Unemployment or economically inactive persons	871	460,540.7	63.0
Education level	Elementary school or below	646	300,058.1	41.1
	Middle school	183	103,172.3	14.1
	High school	334	234,211.6	32.1
	College and over	144	93,089.0	12.7
Household income	Low	485	234.695.7	32.8
	Middle-low	352	217.520.6	30.4
	Middle-high	258	148.186.3	20.7
	High	193	114.889.7	16.1
Residential area	Urban area	975	588,529.7	80.2
	Rural area	340	145,355.0	19.8
Activity restriction	Yes	566	289,376.0	39.5
	No	747	443,782.3	60.5
Subjective health assessment	Poor	227	111,231.7	15.2
	Below average	538	292,871.3	39.9
	Average	383	238,217.8	32.5
	Good	154	85,071.3	11.6
	Excellent	12	6134.4	0.8
Subjective body image	Skinny	237	119,162.7	16.4
	Average	438	238,053.9	32.8
	Obesity	629	369,354.0	50.9

**Table 2 ijerph-17-03502-t002:** Results of the effects of physical activity (e.g., strength exercise, flexibility exercise, walking) on stress in Korean adult women with depressive disorder by logistic regression analysis.

Type of PA	Participation		Perception of Stress				
			OR		95% CI		*p*
Strength Exercise	No		Reference				
	Yes		0.717		0.455–1.131		0.152
Flexibility Exercise	No		Reference				
	Yes		1.434		1.043–1.973		0.027 *
Walking	No		Reference				
	Yes		1.268		0.882–1.825		0.200
Cox & Snell R2:	0.109	Nagelkerke R2:	0.148	McFadden R2:	0.087	Wald F:	8.850 ***

Note: OR = odds ratio, CI = confidence interval, * *p* < 0.05, *** *p* < 0.001. Perception of stress (reference category = 1. Feeling a lot of stress). Adjusted variables: Age, activity restriction, subjective body image, subjective health assessment.

**Table 3 ijerph-17-03502-t003:** Results of the effects of physical activity (e.g., strength exercise, flexibility exercise, walking) on suicidal ideation in adult women with depressive disorder by logistic regression analysis.

Type of PA	Participation		Suicidal Ideation				
			OR		95% CI		*p*
Strength Exercise	No		Reference				
	Yes		1.236		0.786–1.943		0.358
Flexibility Exercise	No		Reference				
	Yes		0.682		0.496–0.937		0.018 *
Walking	No		Reference				
	Yes		1.111		0.758–1.629		0.588
Cox & Snell R2:	0.099	Nagelkerke R2:	0.133	McFadden R2:	0.076	Wald F:	7.513 ***

Note: OR = odds ratio, CI = confidence interval, * *p* < 0.05, *** *p* < 0.001. Suicidal ideation (reference category = 2. No). Adjusted variables: Age, activity restriction, subjective body image, subjective health assessment.

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
