# Peer review of "Effects of Physical Activity on the Stress and Suicidal Ideation in Korean Adult Women with Depressive Disorder"

_ijerph, 2020, doi:10.3390/ijerph17103502_

Round 1

Reviewer 1 Report

The authors have addressed my remaining concerns.

Author Response

The authors have addressed my remaining concerns.

Reply: Thank you for your review.

Reviewer 2 Report

The paper appears very improved with respect with its previous version. Authors have added the several limits of the study in the limitation section, and they have also better explained some methodological procedures. However some negative points still remains

  • The paper need a substantial native English editing
  • Literature shows a lot of very corroborated risk factors for suicidal ideation in subjects with MDD (depressive severity, mixed features, some pharmacological treatment, etc.). in the limitation section of the manuscript, Authors have to really stress that these risk factors were not analyzed in the present study.
  • PA questions explored PA in the “last week”, whereas suicidal ideation was investigated in the last year. This point appears like the major problem of the study that is aimed  at examining the effects of PA on the stress and suicidal ideation

In consideration of the suggestions mentioned above, this study can be accepted for publication with the modifications mentioned above.

Round 2

Reviewer 2 Report

The paper can now be accepted for publication

This manuscript is a resubmission of an earlier submission. The following is a list of the peer review reports and author responses from that submission.

Round 1

Reviewer 1 Report

This cross-sectional survey of Korean adult women examined a sub-sample of depressed women drawn from the Korea National Health and Nutrition Examination Survey to study the association between physical activity (subdivided into strength exercise, flexibility exercise and walking) on stress and suicide ideation. The study found interesting significant associations between flexibility exercise and reductions in stress and suicide ideation in depressed Korean women. Several issues were raised about the current research report:
1. Why didn’t the researchers include a comparison sample of Korean women without depression perhaps matched for age to clarify whether these relationships existing in other community residing women?
2. Why were the activity variables collapse into categorical variables “Never versus participation in walking”? Collapsing the variable into categories may be particularly problematic if “many Korean adults are doing moderate intensity strength exercise and walking…” as indicated in the discussion on page 8.
3. The authors should have discussed why flexibility exercise might be superior to strength exercise and walking as the break down into subtypes of physical activity was a particular strength of the study.
4. Overall the paper was poorly written and many of the sentences were unclear e.g. page 2, line 53, 54 “These stress leads to depression, which may appear as suicidal ideation or suicidal behavior.”
5. The authors should avoid using the phrase “commit suicide” and rather use non-stigmatizing language such as “died by suicide”.

Author Response

We would like to thank reviewer #1 for a thorough review of our manuscript. We believe we have addressed all comments and edits below. In our manuscript, all comments and edits are in blue font. Any common edits (e.g., unclear sentences) from the reviewers are in green font.

Reviewer #1:

  • Why didn’t the researchers include a comparison sample of Korean women without depression perhaps matched for age to clarify whether these relationships existing in other community residing women?

Thank you, lines 316-321 on page 9 have been updated.

  • Why were the activity variables collapse into categorical variables “Never versus participation in walking”? Collapsing the variable into categories may be particularly problematic if “many Korean adults are doing moderate intensity strength exercise and walking…” as indicated in the discussion on page 8.

Thank you, lines 321-325 on page 9 have been updated.

  • The authors should have discussed why flexibility exercise might be superior to strength exercise and walking as the break down into subtypes of physical activity was a particular strength of the study.
    Thank you, lines 274-281 on page 8 have been updated.
  • Overall, the paper was poorly written and many of the sentences were unclear e.g. page 2, line 53, 54 “These stress leads to depression, which may appear as suicidal ideation or suicidal behavior.”

Thank you, lines 55-57 on page 2 have been updated.

  • The authors should avoid using the phrase “commit suicide” and rather use non-stigmatizing language such as “died by suicide”.

Thank you, lines 71 on page 2 has been updated.

Reviewer 2 Report

The present manuscript is aimed at examining the effect of different types of physical activity on stress and suicidal ideation on a large sample of depressed woman in South Korea.

The study is affected by several flaws:

  • The English need to be very improved
  • Depressive disorder need to be changed in “Major Depressive Disorder”. It can be named a “mental disorder”, “mental health problem” or a “psychiatric disorder” but not a psychological disorder
  • Again the direct causality between stress and depression described in the introduction is far from the modern scientific evidence. MDD etiology is still not clear, and many biological, psychological and social factors contribute to its development.
  • The introduction need to be more structured. Furthermore, the majority of the study presented are related to South Korea, reducing the possibility to generalize the study to other country.
  • The size of the study sample need to be clearly described. How MDD was diagnosed in the study sample?
  • The use of not structured and validated scales for stress or suicidal ideation represents an important limit to the study
  • The sample is not homogeneous in terms of several socio- demographic variables showed in table 1. Are they considered for the analysis?
  • Other factors than PA can impact on stress or suicidal ideation (e.g. treatment modalities or severity of depression).
  • Logistic regression model need to be better described (e.g., what about the Cox R2, Nagelkerke R2, Hosmer-Lemeshow test and the Global-goodness-fit percentage ?)
  • A structured limitation section need to be added
  • A causality effect between flexibility PA, stress or suicidal ideation described in the conclusion can’t be achieved by this study, due to its cross-sectional design

In conclusion the present version of the study can’t be accepted for publication.

Round 2

Reviewer 1 Report

The authors have not address moving from continuous variable to categorical variables and this is a may weakness of the paper. There are still many problems with the editing of the English in the manuscript.